# Pt–Ti Alloy Coatings Deposited by DC Magnetron Sputtering: A Potential Current Collector at High Temperature

**Pascal Briois** [1,*] **, Mohammad Arab-Pour-Yazdi** [1] **, Nicolas Martin** [2] **and Alain Billard** [1]

1   FEMTO-ST Institute (UMR CNRS 6174), Univ. Bourgogne Franche-Comté, UTBM, 2 Place Lucien Tharradin, F-25200 Montbéliard Cedex, France; mohammad.arab-pour-yazdi@utbm.fr (M.A.-P.-Y.); alain.billard@utbm.fr (A.B.)
2   FEMTO-ST Institute (UMR CNRS 6174), Univ. Bourgogne Franche-Comté, 15 Avenue des Montboucons, F-25030 Besançon Cedex, France; nicolas.martin@femto-st.fr
*   Correspondence: pascal.briois@utbm.fr; Tel.: +33-3-84-58-37-01; Fax: +33-3-84-58-37-37

**Abstract:** Metallic platinum–titanium coatings were deposited by co-sputtering of two metallic Pt and Ti targets in pure argon atmosphere. The titanium concentrations varied from 0 to 47 atomic percent and were adjusted as a function of the current applied to the titanium target. The structural and chemical features of these films were assessed by X-ray diffraction (XRD) and scanning electron microscopy (SEM). All as-deposited coatings exhibit a perfect covering of the alumina pellets' substrate surface. The coatings containing more than 4 at.% Ti are amorphous, whereas the others crystallize in the face-centered cubic (fcc) structure of platinum. After an annealing treatment under air for 2 h, all of the coatings adopt the fcc structure with a crystallization temperature depending on the titanium content. For titanium concentrations higher than 32 at.%, the $TiO_2$ phase appears during the annealing treatment. For the smaller film thickness of Pt–Ti alloys (15 nm), the Ostwald ripening mechanism is observed by SEM increasing the annealing temperature regardless of the content of Ti. The film resistivity measured at room temperature is lower than $7 \times 10^{-4}$ $\Omega$·cm and increases with the temperature to achieve an insulating behavior (in air and reducing atmosphere $Ar-H_2$ (90-10) at 1123 K the resistivity is $\rho \approx 10^{+36}$ $\Omega$·cm). When the thickness of intermetallic $Pt_3Ti$ layer is higher than 50 nm, the coating is continuous and the resistivity is below $5 \times 10^{-4}$ $\Omega$·cm in air and in reducing atmosphere (Ar with 10% of $H_2$) up to 1273 K.

**Keywords:** resistivity; magnetron sputtering; coating; Pt–Ti alloys

## 1. Introduction

Platinum is an interesting material used as a catalyst [1–4] for fuel cell electrodes [3–5] and a protective layer against industrial corrosive atmosphere [2] or sensors [1,6]. However, this material is very expensive, and its use as thin film is susceptible to minimizing the quantity used. One powerful technique to avoid its waste is DC magnetron sputtering. Kawamura et al. [7] showed that platinum grows with a Volmer–Weber mode. In order to obtain a continuous film, the minimal thickness is around 0.4 nm. However, some problems appear when pure platinum is used, such as pollution by CO during the catalysis process [8], or microstructure coarsening with the temperature that promotes pinholes or discontinuous films [9,10]. Many studies have focused on stabilizing continuous platinum coating with the temperature, and one solution is to deposit metallic platinum alloys (Pt–X): Pt–Ni [11,12], Pt–Ru [8], Pt–Ir [13,14], and Pt–Ti [15–19]. Among the solutions available in the literature, the Pt–Ti system presents some interesting properties, such as high thermal stability, very good behavior as a catalyst, and high resistance against corrosion due to the negative value of the formation enthalpy of

the Pt–Ti bonds [15]. The phase diagram of Pt–Ti alloys firstly presented by Murray [20] was supported by many publications, and some minor corrections have been proposed on this system so far [21].

In this paper, we deposited Pt–Ti metallic thin films through pulsed DC magnetron sputtering with various titanium concentrations. The chemical and morphological features were assessed via scanning electron microscopy equipped with energy dispersive X-Ray spectroscopy. The structural evolution was characterized by X-ray diffraction, and the electrical properties were determined by the four-point probe technique as a function of the temperature under static air and under reducing atmosphere (Ar with 10% of $H_2$). Firstly, we deposited a thin coating within a large panel of titanium contents (0 to 47 at.%). Then, an intermetallic coating $Pt_3Ti$ with different thicknesses was synthesized (10 to 100 nm). The aim of this study is to propose the minimum thickness of $Pt_3Ti$ intermetallic layer required to be used as a current collector (for example in fuel cells) at high temperatures.

## 2. Experimental Details

Coatings have been deposited by co-sputtering of metallic targets in pure argon atmosphere. Pulsed DC supplies power the platinum (Pt, purity 99.95%) and titanium (Ti, purity 99.5%) targets mounted on balanced magnetrons—respectively, an Advanced Energy Pinacle+ pulsed at a fix frequency of 100 kHz for the Pt and 150 kHz for the Ti target. The reactor was a 90-litre cylinder pumped down with a turbo molecular pumping system allowing a base vacuum of less than $10^{-4}$ Pa before refilling with argon at a convenient pressure. The argon flow rate introduced in the deposition chamber was controlled by a Brooks flowmeter, and the total pressure was measured using an MKS Baratron Gauge (model 690A01TRC, MKS Instrument France SA, Munchen, Germany) [22].

The targets, 50 mm in diameter and 3 mm in thickness, are parallel to the substrate-holder and are spaced 120 mm from each other. Their distance from the substrate holder ($D_{T-S}$) was fixed at 60 mm. Different substrates, such as glass slides and dense alumina pellets (Keral 99, diameter = 16 mm, thickness = 0.63 mm), produced by Kerafol Gmbh (Eschenbach in der Operpfalz, Germany), were positioned at a distance around 55 mm from the axis of the rotating substrate-holder. Alumina substrates were used to carry out the heat treatments as well as the structural, microstructural, and electrical characterizations of the films. Glass slides were used to measure thickness through tactile profiling in addition to the thickness obtained by SEM. During the deposition stage, the different substrates were regularly rotated (20 Round Per Minute). The Pt discharge current was fixed at 0.2 A for all experiments, and the discharge current applied to the Ti target was varied from 0 to 1 A in order to obtain Pt–Ti coatings with different Ti contents. The main sputtering parameters are summarized in Table 1.

**Table 1.** Main deposition conditions of the study.

| | | | |
|---|---|---|---|
| **Air Flow Rate (sccm)** | 50 | **Discharge Current on Pt (A)** | 0.2 |
| **Total Pressure (Pa)** | 0.5 | **Pulse Frequency (kHz)** | 100 |
| | | $T_{off}$ **(µs)** | 2 |
| **Sputtering Time (min)** | 2.5 | **Discharge Current on Ti (A)** | $0 \rightarrow 1$ |
| **Drawing Distance (mm)** | 60 | **Pulse Frequency (kHz)** | 150 |
| | | $T_{off}$ **(µs)** | 3.3 |

The structural features of the coatings were determined through X-ray diffraction in a grazing incidence geometry using an incidence angle of 4 degrees configuration. A BRUKER D8 (Karlsruhe, Germany) focus diffractometer (Co $K_{\alpha1+\alpha2\ radiations}$, $\lambda$ = 0.178897 nm) equipped with a LynxEye linear detector (Bruker, Karlsruhe, Germany) with a fix incidence of 4° was used. Diffractograms were collected under air flow for 10 min in the [20°–80°] scattering angle range by steps of 0.019°. The morphology of the coatings was characterized via scanning electron microscopy (FEG SEM) using a JEOL JSM 7800F (Croissy sur Seine, France) equipped with energy-dispersive X-ray spectroscopy (EDS, XFlash 6|30, Bruker nano, Berlin, Germany) for chemical measurements. Resistivity measurements were first made at room temperature by means of a certified Jandel device (Multi height probe,

Jandel Engineering Limited, Lindslade, UK), which allows the determination of the form factor of the measured cell temperature. Then, the electrical resistivity measurements as a function of temperature were taken with an HP 3458A multimeter (Agilent, Massy, France) on the Pt–Ti film deposited on alumina substrates. The four-point probe technique with four Pt aligned electrodes was employed. The two outer probes are the current-carrying electrodes (I1, I2), and the two inner ones were used to measure the voltage (E1, E2). The cell was placed inside an alumina tube and positioned in a Pekly furnace. The resistivities of the films were recorded from room temperature up to 1273 K with temperature steps and stabilization times of about 25 K and 20 min, respectively. Measurements were taken under static air and under reducing atmosphere (total flow rate was 60 sccm, and the gas was composed by Ar with 10% of $H_2$).

## 3. Results and Discussion

### 3.1. Influence of Ti Content

Figure 1 shows the titanium content in the coating as a function of the current applied to the titanium target. The linear evolution was assumed to proceed from the metallic flux increasing quite linearly with the discharge current on the Ti target in neutral atmosphere. Regardless of the film composition, as-deposited Pt–Ti metallic films perfectly cover the surface of alumina substrates. The micrographs obtained via SEM of the film top-view (Figure 2) show that all the samples perfectly reproduce the substrate surface topography. Only the coatings of pure platinum and with 4 at.% titanium are crystallized in the face-centered cubic structure of platinum. All other films with higher Ti content present an amorphous structure. XRD patterns obtained after two hours of thermal oxidation at 1173 K in air of the Pt–Ti metallic films are shown in Figure 3. All coatings are crystallized under the fcc structure according to the pure platinum structure. In comparison with the cubic cell of bulk platinum, the cell is more and more deformed with each increase in titanium content. The atomic radii are 0.139 and 0.147 nm for the platinum and titanium, respectively. Basically, the substitution of Pt atoms by titanium increases the lattice constant (i.e., the Bragg angle corresponding to (111) planes of platinum moves to smaller values while increasing the titanium content in the coating). Figure 3 shows a reverse behavior for titanium concentration range from 4 at.% to 25 at.%. Spencer [17] explained this phenomenon through the attractive bond length between platinum and titanium, which implies that the Pt–Ti bond is shorter than the Pt–Pt bond. Irrespective of the Ti content, the coatings remain a supersaturated fcc solid solution after annealing treatment at 1173 K. The intermetallic phases $Pt_3Ti$, $Pt_5Ti_3$, and PtTi of the Pt–Ti system predicted by Biggs et al. [21] do not appear in these coatings.

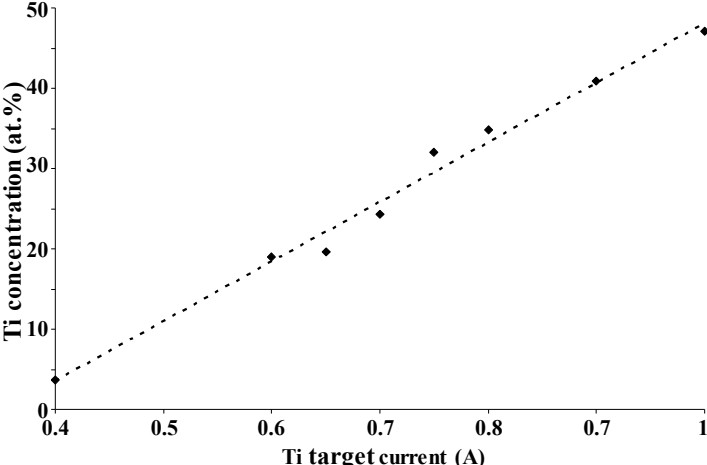

**Figure 1.** Titanium concentration measured by EDS as a function of the discharge current applied on the Ti target (The intensity on platinum is fixed at 0.2 A).

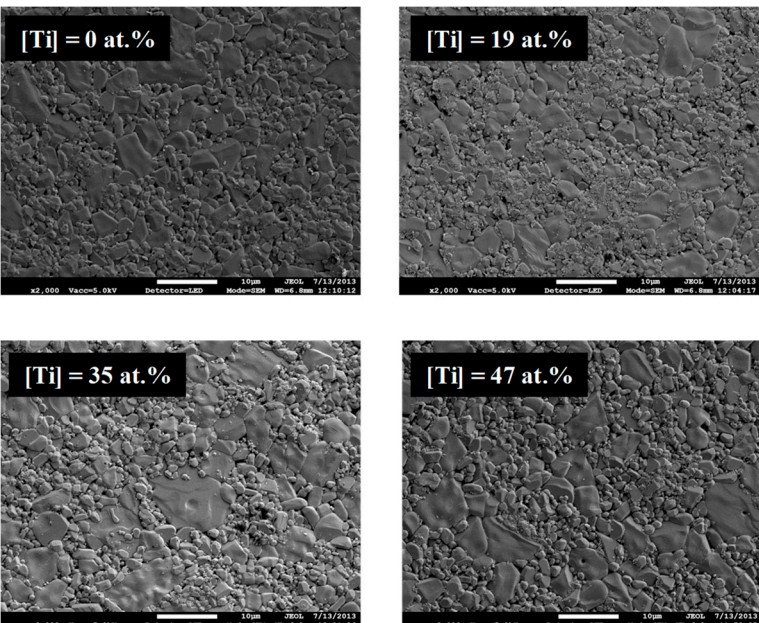

**Figure 2.** SEM observations of the top surface of Pt–Ti coatings with different Ti concentrations as-deposited on alumina pellets.

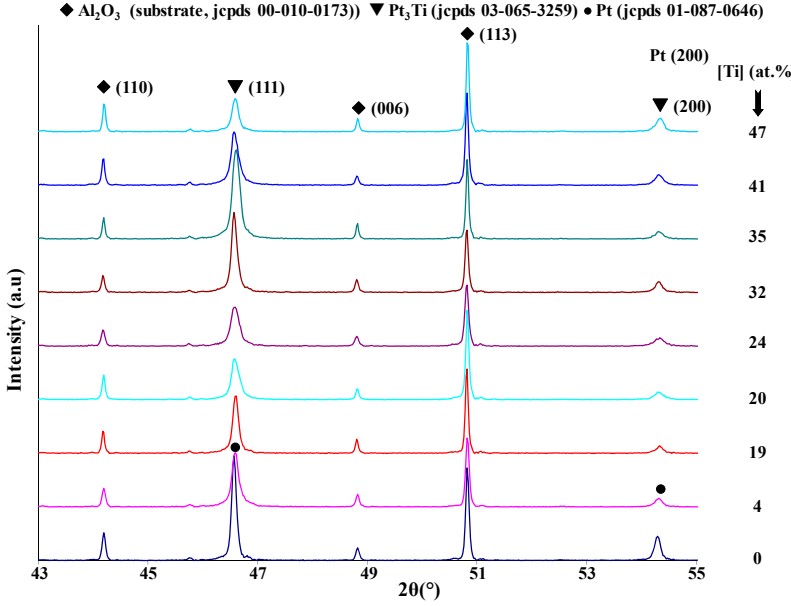

**Figure 3.** XRD patterns of the Pt–Ti films deposited on alumina pellets after annealing treatment in air at 1173 K during two hours and for different titanium concentrations.

Figure 4 represents the resistivity of around 15 nm thick platinum-based coating of different Ti contents as a function of annealing temperature. The resistivity is quite constant when the temperature is lower than 673 K. As the temperature rises, the film resistivity increases as well [9,10]. The transition temperature from conductor to insulator is a function of the titanium content. Figure 5 shows the XRD patterns of four coatings with different titanium concentrations as a function of the annealing temperature. The temperature at which the films start to crystallize depends on the Ti concentration. It extends over a range from 673 to 873 K, respectively, for pure Pt and Pt with 47 at.% of Ti. In the case of pure Pt and Pt with 4 at.% of Ti, the films are crystallized, roughly worked up, under the face-centered cubic structure. For films containing from 19 at.% to 47 at.% of Ti, crystallization begins

from 873 K. Some jumps on the electrical resistivity value are observed (Figure 4). The temperature of these jumps is expected to correspond to the morphological evolution of the coatings. Indeed, SEM observation performed after electrical measurements up to 1273 K clearly shows the discontinuity of the films with the lower Ti contents that consist in isolated islands of submicronic size. For higher Ti contents, the film morphology appears granular, and although it seems to still percolate, the high resistivity is assumed to proceed from a strong refinement of the layer between the grains.

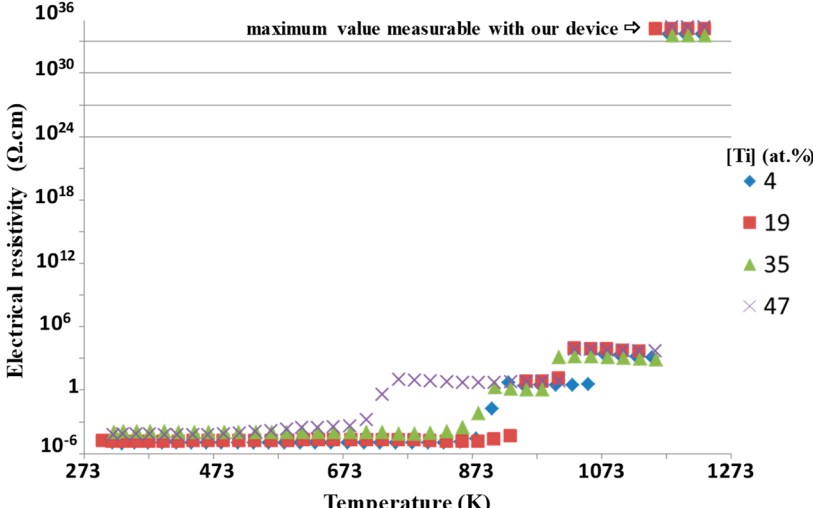

**Figure 4.** Resistivity measurements of different Pt–Ti coatings as a function of the temperature and for four Ti concentrations.

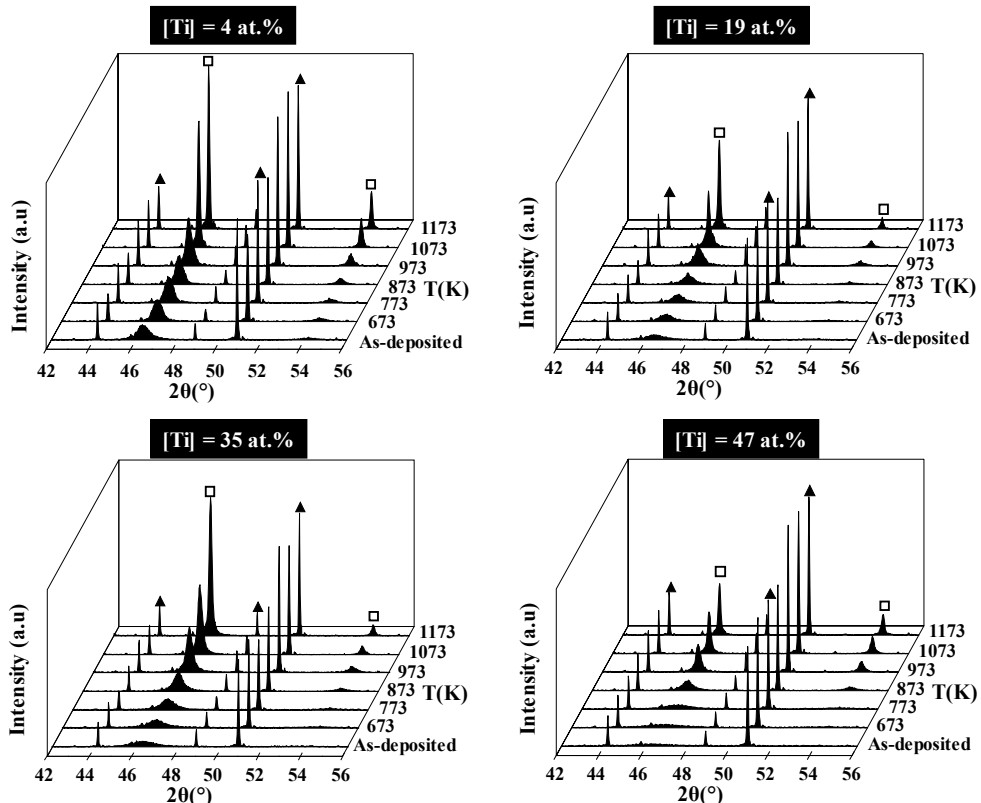

**Figure 5.** XRD patterns as function of the annealing treatment for different titanium concentrations ($\square$ $Pt_3Ti$, $\blacktriangle$ alumina).

It is worth noting that the continuity of platinum coating with the temperature is very important to use this material as a current collector for fuel cell applications. Figure 6 shows the top-view observation via SEM of the films after electrical measurements performed at 1273 K for coatings with four different concentrations of titanium (4 at.%, 19 at.%, 35 at.%, 47 at.%). When the temperature is higher than 873 K, the coatings containing up to 19 at.% Ti are not continuous and are composed by small islands a few nanometers of platinum on the alumina surface [9,10,23]. This surface morphology is attributed to the Ostwald ripening mechanism during the sintering of the film with the annealing treatment during the electrical measurement [10]. The coating containing 35 at.% Ti also presents less covering of the alumina substrate, but it still percolates, thus maintaining a measurable resistivity. Finally, the coating containing 47 at.% Ti remains continuous.

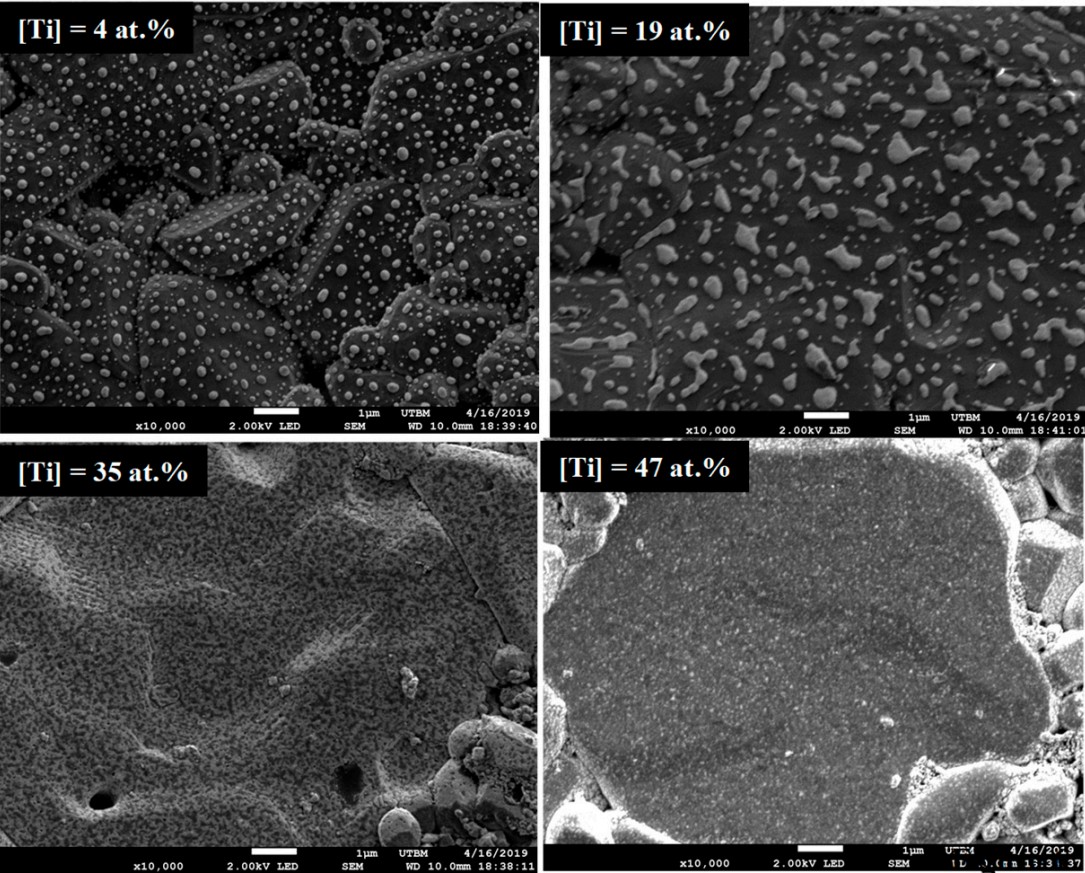

**Figure 6.** SEM micrographs of the top surface of different Pt–Ti films after resistivity measurements.

### 3.2. Influence of the Thickness of $Pt_3Ti$ Coatings

The convenient composition of $Pt_3Ti$ is obtained if the discharge currents are 0.2 A pulsed at 100 kHz on the Pt target and 0.7 A pulsed at 150 kHz on the Ti target. The coatings were deposited under 0.5 Pa argon pressure, and the sputtering time was adjusted to synthesize the films with different thicknesses (26, 52, and 104 nm). The thinner film annealed under air presents an insulating behavior over about 1173 K, while thicker films continue to conduct up to about 1273 K (Figure 7a). When the annealing has been realized under reducing atmosphere, an insulating behavior is observed over about 1100 K for the 26 nm thick film and over about 1200 K for the 52 nm thick one. The thickest film behaves as a conductor for up to at least 1273 K (Figure 7b). In all atmospheres, the resistivity decreases with increasing the film thickness and remains stable up to at least 1273 K for the highest thickness ($\approx$ 104 nm). This behavior is consistent with the SEM observation (top-view in Figure 8), where the thinnest film consists in separated islands, whereas the thicker one still percolates under

air and remains quite continuous under reducing atmosphere. The benefit effect of titanium addition against grain coarsening and the Ostwald ripening mechanism is less important when the thickness increases. A minimum of 50 and 100 nm thickness is necessary to obtain a stable current collector at high temperature application and avoid some island formation under air or under reducing atmosphere.

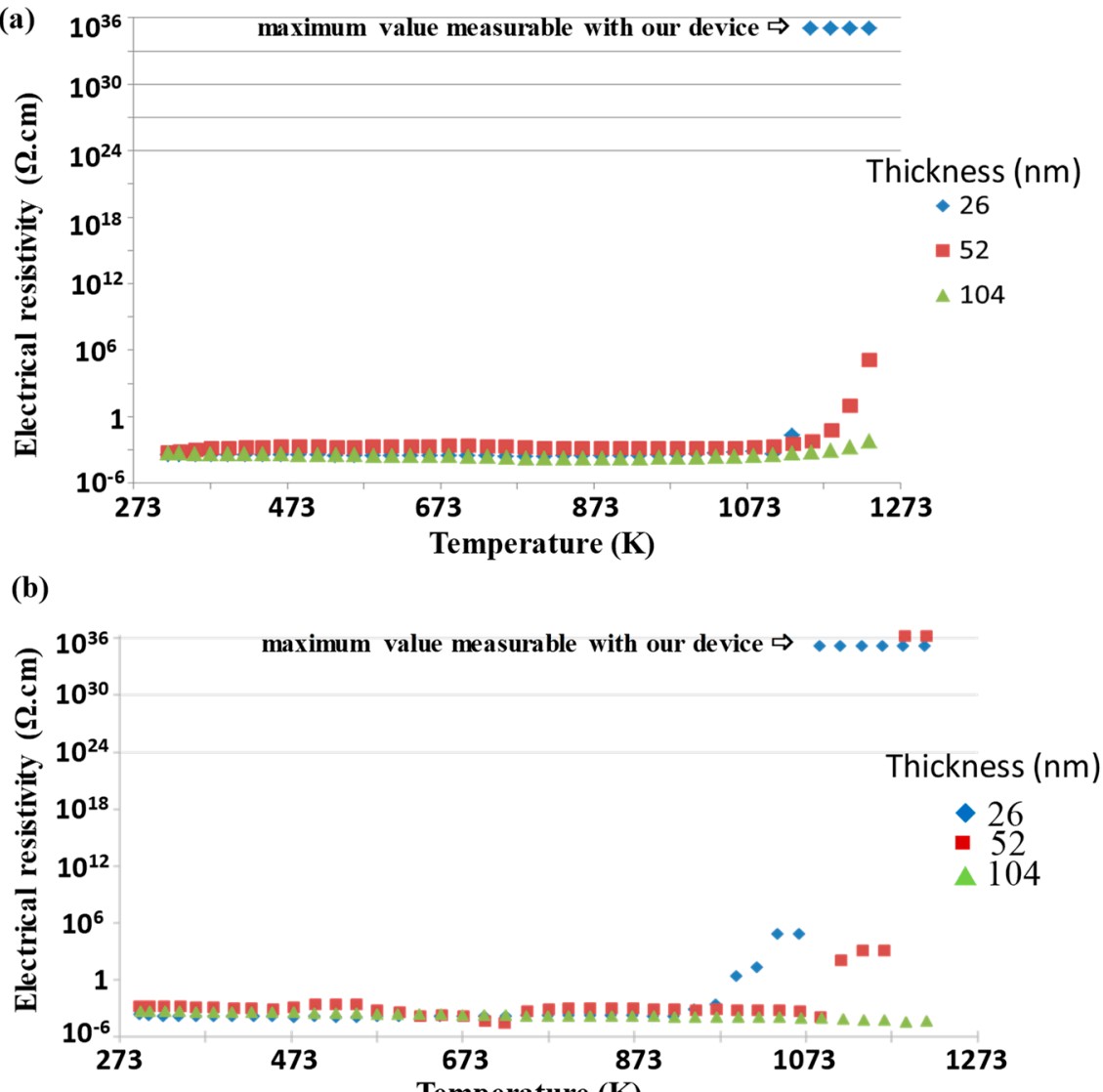

**Figure 7.** Resistivity measurements of $Pt_3Ti$ coatings as a function of the temperature and for three thicknesses (**a**) under static air and (**b**) under reductive atmosphere composed of 90% argon and 10% hydrogen.

**air**　　　　　　　　**reducing atmosphere**

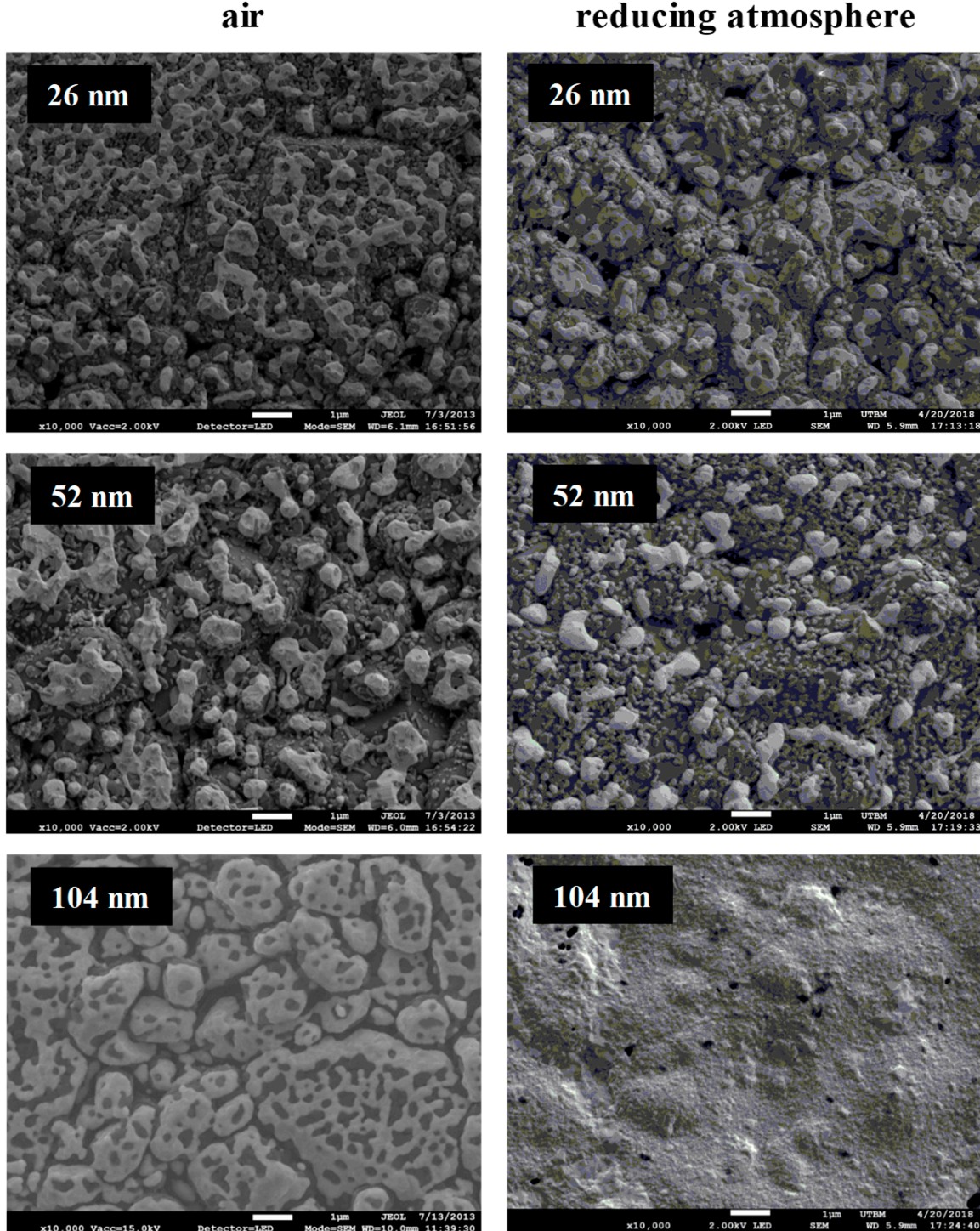

**Figure 8.** SEM micrographs of the top surface of Pt₃Ti coatings with different thicknesses after the resistivity measurements under air and under reductive atmosphere at 1273 K.

## 4. Conclusions

Platinum–titanium is a potential material as a current collector in the fuel cell applications. Pt–Ti thin films were elaborated via co-sputtering of two metallic Pt and Ti targets in pure Ar atmosphere. The titanium concentration of coating ranged as a function of intensity applied on the titanium target. All as-deposited coatings were perfectly covering and reproduced the substrate surface morphology of alumina pellets. X-ray diffraction results showed that the as-deposited metallic films with or without

small addition of titanium are crystallized in the fcc platinum structure, while for Ti concentrations higher than 4 at.%, the coatings become amorphous. For the thinner coating (15 nm), the Ostwald ripening phenomenon was observed with the annealing treatment irrespective of the Ti concentration.

We also observed that if the $Pt_3Ti$ coating thickness is higher than 100 nm, the electrical and morphological properties as a function of the annealing treatment remain stable. This film could be a good potential candidate as a current collector at high temperature under oxidizing or reducing atmosphere.

**Author Contributions:** Conceptualization, M.A.-P.-Y., P.B. and A.B.; methodology, M.A.-P.-Y.; validation, A.B.; formal analysis, M.A.-P.-Y., P.B.; investigation, M.A.-P.-Y., P.B. and A.B.; resources, A.B.; data curation, M.A.-P.-Y., P.B.; writing—original draft preparation, P.B. and M.A.-P.-Y.; writing—review and editing, M.A.-P.-Y., P.B., N.M. and A.B.; supervision, A.B.; project administration, P.B. and A.B.; All authors have read and agreed to the published version of the manuscript.

**Funding:** The authors are indebted to the Pays de Montbéliard Agglomération (PMA) for its financial support.

**Acknowledgments:** This work was done within the FEMTO-ST SURFACE platform.

**Conflicts of Interest:** The authors declare no conflict of interest.

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
