# Peer review of "Pt–Ti Alloy Coatings Deposited by DC Magnetron Sputtering: A Potential Current Collector at High Temperature"

_coatings, doi:10.3390/coatings10030224_

Round 1

Reviewer 1 Report

The publication contains interesting results concerning Pt-Ti films deposited by magnetron sputtering method. The authors noted that if the thickness of Pt3Ti coating is higher than 100 nm, the electrical and morphological properties of the annealing function remain stable. This indicates the possibility of using such films as a high temperature current collector in oxidizing or reducing atmosphere.

The manuscript is well constructed but also contains some errors and requires some corrections and additions, for example:

1. Line 67: For which tests were glass slide substrates used?

2. Line 73: No Table 1. To be completed.

3. Lines 98-99: The statement: "Only the coatings of pure platinum and with 4 at.% Titanium are crystallized in the face-centered cubic structure of platinum." - - This is not apparent from SEM research and requires more precise explanation.

4. Line 116. For comparison, the surface image of alumina pellets substrates is missing before the production of Pt-Ti films.

5. Line 119: Figure 3 – XRD patterns of Pt-Ti films before the annealing could be useful for comparison.

6. Lines 127-128: The statement "The crystallization temperature slightly increases with the titanium concentration." requires more precise explanation with reference to Figure 5.

7. Lines 138-139: There are no visible platinum peaks on the XRD patterns (Figure 5), so a "• Pt" marker is not needed.

8. Lines 150, 175 and 116: SEM micrographs of film surfaces (in Figures 1, 6 and 8) should be at the same magnification to be able to compare the size of visible Pt-Ti particles. Additionally, EDS mapping images of the surface of these films could show the distribution of elements and coating discontinuities.

In general, the manuscript is acceptable after necessary additions and corrections.

Author Response

Dear Reviewer 1,

we thank you for the work you have done for the revision of our article, we have modified the text (inserted in red in the text) according to your comments and we have answered your questions in red.

please find our answer to all of your comments

1. Line 67: For which tests were glass slide substrates used?

To measure the thickness by profilometer in order to have two measurements to determine it

2. Line 73: No Table 1. To be completed.

It is a mistake, we have inserted table 1

3. Lines 98-99: The statement: "Only the coatings of pure platinum and with 4 at.% Titanium are crystallized in the face-centered cubic structure of platinum." - - This is not apparent from SEM research and requires more precise explanation.

The sentence you quote is related to DRX and not the morphology observed by SEM

4. Line 116. For comparison, the surface image of alumina pellets substrates is missing before the production of Pt-Ti films.

When producing a thin film (<500 nm) by PVD, the morphology of the substrate is always revealed, the substrates being fried alumina and not mirror polished before deposition, the granular morphology of the substrate is observed.

5. Line 119: Figure 3 – XRD patterns of Pt-Ti films before the annealing could be useful for comparison.

We did not want to put all the XRD of the as-deposited films for the different compositions so as not to have too much image, but we did it in Figure 5 for the four relevant compositions. (raw and after annealing at different temperatures)

6. Lines 127-128: The statement "The crystallization temperature slightly increases with the titanium concentration." requires more precise explanation with reference to Figure 5.

We have modified this part in red in the text

7. Lines 138-139: There are no visible platinum peaks on the XRD patterns (Figure 5), so a "• Pt" marker is not needed.

Thank you for this remark, we removed it from the legend

8. Lines 150, 175 and 116: SEM micrographs of film surfaces (in Figures 1, 6 and 8) should be at the same magnification to be able to compare the size of visible Pt-Ti particles. Additionally, EDS mapping images of the surface of these films could show the distribution of elements and coating discontinuities.

Each image of the same figure is at the same magnification, we have chosen to have magnifications by different figure in order to better observe the phenomenon of coalescence. We had tried to make EDS mapping images however the results were not relevant because we did not metallize the surface to be sure to observe the layers alone

Best regards

Pascal Briois

Reviewer 2 Report

1. Line 26, Use “x” instead of asterisk (*)

Experimental details: the following information must be provided: magnetron voltage and power, gas flow rate and gas pressure during operation,   

2. Line 73: where is table 1 (missing)?

3. Line 75: Co K-alpha can be written on 1 line, provide wavelength

4. Line 80: what does it mean, in particular, how does a certified (?) device allow for a determination of the “form factor of the measured cell temperature”.

5. Line 111 and Figs. 3 and 5: the Pt3Ti phase does not appear in these coatings. In Fig. 3 Pt (111) and Pt(200) are indicated while the legend refers to Pt3Ti. Where do reflections from Pt and/or Pt3Ti lattice appear and which ones are observed. Likewise, legends of Fig. 5 refer to Pt and Pt3Ti while only Pt3Ti (blacktriangle) but no Pt (blackcircle) pattern are noted.    

6. Fig. 1, caption: Delete “Evolution of”.

7. Line 154/156: where is the “convenient” composition of Pt3Ti displayed? Where in the Experimental section did you mention a “pulsed” discharge? What are the exact conditions (pulse width, duty cycle, discharge voltage and power, gas pressure?   

8. Figs. 7: Is there a difference if resistivity is measured during temperature increase or decrease (after heating to 1273 K)?

9. Fig. 8: what is the size of the displayed area? Micrographs were taken “after” the resistivity measurements which implies, according to my understanding, that the film were heated to 1273 K. If so, this should be mentioned. Do you have micrographs taken without heating?

Author Response

Dear Reviewer 2,

we thank you for the work you have done for the revision of our article, we have modified the text (inserted in red in the text) according to your comments and we have answered your questions in red.

1. Line 26, Use “x” instead of asterisk (*)

We have replaced the terms

Experimental details: the following information must be provided: magnetron voltage and power, gas flow rate and gas pressure during operation,  

2. Line 73: where is table 1 (missing)?

We have inserted Table 1 that we forgot, and all the information on the summary is included

3. Line 75: Co K-alpha can be written on 1 line, provide wavelength

We have inserted the wavelength in the red text

4. Line 80: what does it mean, in particular, how does a certified (?) device allow for a determination of the “form factor of the measured cell temperature”.

Our low temperature device has a certificate of analysis which tells us the distance between each probe and we have a table that gives us the form factor. The high temperature cell was produced by a laboratory and we do not know very precisely the distance between the Pt wires; To make our measurements, we make a first measurement with our certified cell and we compare the resistance value with that obtained with our high temperature cell. This experience allows us to determine the geometric factor of our cell for each test.

5. Line 111 and Figs. 3 and 5: the Pt3Ti phase does not appear in these coatings. In Fig. 3 Pt (111) and Pt(200) are indicated while the legend refers to Pt3Ti. Where do reflections from Pt and/or Pt3Ti lattice appear and which ones are observed. Likewise, legends of Fig. 5 refer to Pt and Pt3Ti while only Pt3Ti (blacktriangle) but no Pt (blackcircle) pattern are noted.  

The diffraction peaks of Pt and Pt3Ti have the same diffraction angles, following the request of another reviewer we removed from legend the reference of Pt

6. Fig. 1, caption: Delete “Evolution of”.

We removed the term "evolution"

7. Line 154/156: where is the “convenient” composition of Pt3Ti displayed? Where in the Experimental section did you mention a “pulsed” discharge? What are the exact conditions (pulse width, duty cycle, discharge voltage and power, gas pressure?  

All the information you want is in table 1 which we forgot to put when laying out in Coatings template

8. Figs. 7: Is there a difference if resistivity is measured during temperature increase or decrease (after heating to 1273 K)?

Yes there would be a difference, during the rise the film coaslece gradually which induces an increase in the resisvity. If we had measured during the descent, the film would be completely insulating

9. Fig. 8: what is the size of the displayed area? Micrographs were taken “after” the resistivity measurements which implies, according to my understanding, that the film were heated to 1273 K. If so, this should be mentioned. Do you have micrographs taken without heating?

The morphology of the raw production films is the same as in FIG. 2 (covering and adherent), an image was not inserted without heating since this brought nothing more in comparison with FIG. 2.

We have specified the temperature in the legend of figure 8

Best regards,

Pascal BRIOIS

Reviewer 3 Report

The paper is devoted to the fabrication and characterization of several metallic-titanium coatings which can be used as potential candidates for current collectors operated at high temperatures. The films were obtained by using the non-reactive magnetron sputtering technique and were deposited on glass slides and dense alumina pellets. The simultaneously co-sputtering from platinum and titanium targets allowed the achievement of thin films with various titanium concentrations. The fcc platinum structures were obtained for as-deposited metallic films with Ti concentrations smaller than 4 at. %, while the films revealed to be amorphous for Ti concentrations ranging from 4 to 47 at. %. Also, the influence of Pt3Ti thickness as a function of the annealing treatment was discussed in terms of electrical resistivity and SEM micrographs.   Although the objectives of this work seem to be interesting, minor parts of the manuscript should be improved. My comments are following:

1. Line 67: Please add more details about the reason of using dense alumina pellets as substrates.

2. Line 73: Table 1 is missing.

3. Line 95: It is not clear how many samples were obtained.

4. Line 97: What do you mean by “perfectly”?

5. Lines 62, 69, 77, 143-145: Small English improvements are required.

Conclusively, the study is a well-conducted research and deserves to be published in Coatings journal.

Author Response

Dear Reviewer 3,

we thank you for the work you have done for the revision of our article, we have modified the text (inserted in red in the text) according to your comments and we have answered your questions in red.

1. Line 67: Please add more details about the reason of using dense alumina pellets as substrates.

We use this type of substrate when we want to make temperature measurements because they are stable, insulating, and inexpensive. A dense substrate allows us to make deposits and polish them beforehand

2. Line 73: Table 1 is missing.

We have inserted table 1

3. Line 95: It is not clear how many samples were obtained.

We placed around 10 samples per experiment, and we take 3 per experiment on which we carried out 5 measures the composition then we carry out an average

4. Line 97: What do you mean by “perfectly”?

We would like to say that the film reproduces the surface of the substrate and that it completely covers it

5. Lines 62, 69, 77, 143-145: Small English improvements are required.

We corrected the sentences cited above.

Best regards,

Pascal Briois

Round 2

Reviewer 1 Report

Thank you for clarifying and taking into account my remarks and suggestions.

Generally, a manuscript is acceptable in this form after minor corrections.

Below are some more comments:

1. Line 67: Thank you for explaining to me but the text still does not specify what glass substrates were used for.

2. Line 233: In literature cited No. 13, complete pages 188-191.

Best regards,

Author Response

Dear reviewer 1,

we have inserted the use of different substrates in the text, and completed reference 13.
Thank you for your different comments, and for having revised our article.

Have a nice days

Best regards,

Pascal Briois

Reviewer 2 Report

1. Figure 3 still shows Pt (peaks) and Pt3Ti (legend). Please correct. 

Author Response

Dear Reviewer 2, 

we modified figure 3 as you suggested,
thank you for your comments and for revising our article

Have a nice days,

Best regards

Pascal Briois